# DexSkin: High-Coverage Conformable Robotic Skin for Learning Contact-Rich Manipulation

**Abstract:** Human skin offers a rich tactile sensing stream over large, contoured areas, but replicating this in robotics remains challenging. We present DexSkin, a soft, conformable capacitive skin that provides localized, calibratable sensing and adapts to varied geometries. Integrated on gripper fingers with near-full coverage, DexSkin enables learning-based manipulation, transfers across sensor instances via calibration, and supports real-world online RL. *This work has been published at CoRL 2025 (main track).*

**Keywords:** tactile sensing, contact-rich manipulation

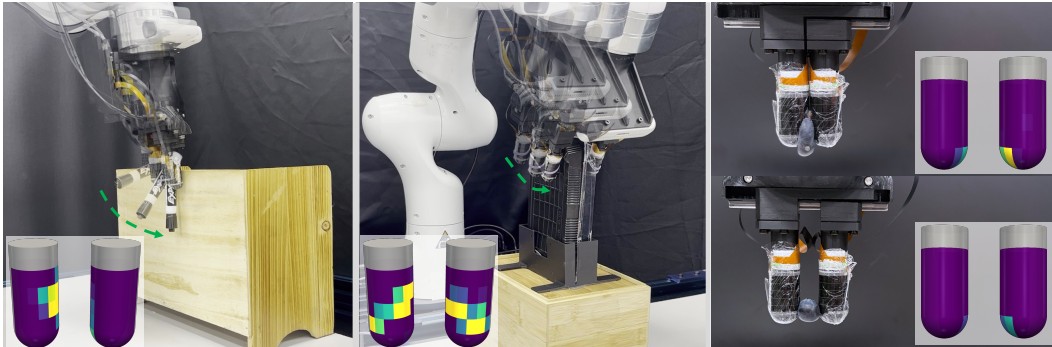

Figure 1: **DexSkin sensors enable contact-rich robotic manipulation policies.** We demonstrate the applications of DexSkin in in-hand pen reorientation, box packaging, and berry transporting tasks. In each frame, we visualize tactile readings from DexSkin as heatmaps, where brighter readings correspond to larger forces. DexSkin's high-coverage sensing capabilities enable it to provide contact signals for a variety of robotic manipulation tasks. At the same time, it possesses desirable properties for learning-based manipulation systems.

## 1 Introduction

Tactile feedback is essential for dexterous manipulation, yet most robotic sensors remain rigid, low-coverage, and poorly adaptable to complex surfaces. We introduce DexSkin, a soft capacitive skin that conforms to end-effector geometry, offers high-density localized sensing, and can be calibrated across hardware instances. Each taxel is individually addressable, allowing localization of contacts across surfaces, and the sensor can be calibrated for consistent readings across hardware instances.

In this work, we describe DexSkin 's design and integration with a robotic fingertip, then evaluate its utility for robot learning. We test whether DexSkin's coverage expands learnable manipulation tasks, assess calibration for model transfer, and demonstrate real-world reinforcement learning on a delicate object picking task. The results highlight DexSkin 's practicality for contact-rich, data-driven manipulation.

## 2 Related Work

**Tactile sensing in robotics.** Robotic tactile sensors generally fall into three categories: vision-based, magnetic, and electrically addressable. Vision-based sensors such as GelSight [1, 2], GelSlim [3], DIGIT [4], and DenseTact [5, 6] capture elastomer deformations with cameras, but are bulky and difficult to miniaturize [7]. Magnetic sensors such as uSkin [8], ReSkin [9], and AnySkin [10]

measure flux changes, but provide sparse signals and require data-heavy calibration. Electrically addressable sensors include resistive [11, 12], impedance (e.g., BioTac [13]), and capacitive [14, 15, 16, 17]. Capacitive designs offer high sensitivity and low power consumption but often rely on costly microfabrication. Our approach also employs a capacitive mechanism, but enables conformable coverage and low-cost, rapid prototyping with individually addressable taxels.

**Learning robotic manipulation with tactile sensors.** Tactile sensing has been integrated into learning robotic manipulation via reinforcement learning [18, 19, 20], predictive models [21, 22, 23, 4], and imitation learning [24, 25, 26, 27]. Since simulating contacts is difficult [28, 29, 30, 31], many works rely on real-world data. We follow this paradigm, focusing on tasks requiring broad coverage and simultaneous contacts.

# 3 The DexSkin Framework

We introduce the main components of the DexSkin framework. While applicable to many morphologies, we focus on fingertip-shaped gripper jaws for a parallel gripper, shown in Figure 2.

**Low-cost, high-performance, conformable DexSkin sensors.** Electrically addressable tactile sensors remain rare in robotics [32, 26] due to sensitivity, cost, and integration challenges. DexSkin is a stretchable capacitive sheet fabricated for under $10 per pair (at 1,000 units) using accessible tools. It provides nearly full-finger coverage, high sensitivity, and individually addressable taxels that can resolve multiple simultaneous contacts.

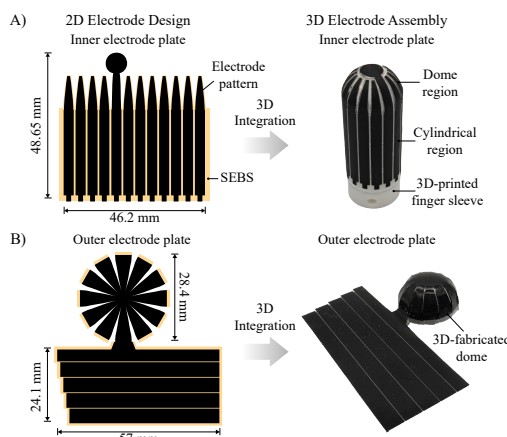

**Tailorable fabrication.** Fabrication begins with computer-aided electrode pattern design, which determines taxel layout and size. The electrode layers are deposited onto a thin elastomer substrate and sealed into a flexible sheet. This single continuous piece can be conformed around both hemispherical domes and cylindrical finger surfaces, yielding dense tactile coverage (Figure 2).

Figure 2: **Designs of electrode patterns.** The 2D pattern design of the inner plate (top left) and outer plate (bottom left) electrodes on the soft SEBS substrate and its appearance after its vertical traces have been conformed onto the dome and cylindrical circumference of the finger sleeve (top right and bottom right). Note that SEBS is transparent and can be hard to distinguish in the post-assembly photo (right).

*We are committed to open-sourcing the detailed fabrication instructions and materials for DexSkin.*

# 4 Evaluating DexSkin for Learning Robotic Manipulation

We evaluate DexSkin for learning robotic manipulation from the following perspectives:

1. Can DexSkin's coverage and tailorability enable robots to learn a range of manipulation tasks?
2. Can calibrating DexSkin allow learned policies to be transferred across sensor instances?
3. Is DexSkin suitable for applications to learning robot behaviors *online*?

## 4.1 Learning Manipulation with Expanded Coverage and Tailorability

We test whether DexSkin 's coverage and resolution improve learned manipulation through two tasks. Policies are trained from 50 teleoperated demonstrations using GELLO [33] and diffusion policies [34, 35]. The first task is in-hand pen reorientation, where the robot must rely on tactile and proprioceptive feedback to reorient a pen and remain robust to human perturbations. The second is box packaging, where the robot must secure a container lid with an elastic band that may be intact

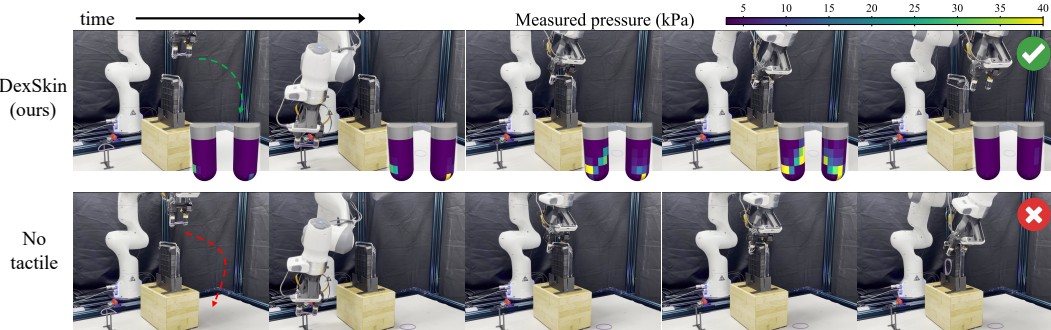

Figure 3: Examples of box packaging task rollouts. *(Top)* **With an initially provided perforated band, the DexSkin policy correctly detects and discards the perforated band based on the tactile reading (col. 1), retrieves the backup (col. 2), and wraps it around the box (cols. 3-5).** We visualize DexSkin sensor readings as if the viewer is facing the robot, looking at the gripper. *(Bottom)* **With an initially provided *un*perforated band, the policy without tactile information unnecessarily replaces the band (cols. 1-2) and then fails to package the box as the band slips off the gripper fingers (cols. 3-5).**

| Model | Pen reorientation | | Box: Non-Perforated Band | | Box: Perforated Band | |
| --- | --- | --- | --- | --- | --- | --- |
| | No perturb | Perturb | Select | Wrap | Select | Wrap |
| No tactile | **19/20** | 0/20 | 0/20 | 0/20 | **19/20** | 6/20 |
| DIGIT [4] | **19/20** | 0/20 | 20/20* | 20/20* | 0/20 | 0/20 |
| DexSkin (ablated) | 12/20 | 0/20 | **19/20** | 14/20 | 1/20 | 0/20 |
| DexSkin (ours) | **19/20** | **19/20** | 18/20 | **17/20** | **19/20** | **15/20** |

Table 1: Pen reorientation and box packaging success rates. Policies with full DexSkin information achieve the most consistent performance across tasks requiring high sensing coverage. For box packaging, we report success in two cumulative stages for evaluation only: (1) determining the correct elastic band to use and (2) physically wrapping the band around the box. *Placing the band around DIGIT's bulkier geometry elongates the band more, making wrapping it around the box easier.

or perforated, requiring it to detect band properties through dorsal fingertip sensing and adapt its behavior accordingly.

**Comparative evaluation.** To test our hypothesis, we compare the following settings:

- *DexSkin (ours)*: Uses the full 120 taxels of DexSkin readings, proprioception, and wrist camera RGB images. Note wrist camera images are provided for the box task only.
- *No tactile*: A baseline identical to *DexSkin (ours)* but excluding all tactile information.
- Ablations include spatially pooled readings for the pen task (*DexSkin (spatial pooling)*), mimicking low-resolution sensors (eg. load cells), and inner-column readings for the box task (*DexSkin (inner cols. only)*), approximating flat sensors. These test the importance of DexSkin 's resolution and coverage.
- *DIGIT*: Uses a pair of DIGIT [4] sensors as gripper fingers and as input to the policy, comparing our system to existing and commercially available sensors.

**Pen task results.** We test two settings: the training setup and one with human perturbation that rotates the pen back to horizontal. Results (Table 1) show most policies succeed without perturbation, but only the full DexSkin policy maintains success under disturbance. The no-tactile baseline repeats the same motion regardless, spatial pooling often fails to estimate pose even unperturbed, and DIGIT misses many contacts outside its limited sensing region.

**Box packaging results.** As shown in Table 1 and Figure 3, only the full DexSkin policy consistently succeeds with both intact and perforated bands. Other policies adopt fixed strategies, misusing bands that are visually indistinguishable, while DexSkin correctly selects based on perceived tension. Full DexSkin input also improves wrapping success, completing 86% of rollouts versus 32% (No tactile) and 70% (inner cols. only).

## 4.2 Calibration and Model Transfer Across Sensor Instances

Tactile sensors are difficult to manufacture consistently, as optical, magnetic, and resistive designs often produce signal variations across units [10, 36]. Because learned models are highly sensitive to such shifts, replacing sensors can render prior training unusable. We introduce a calibration procedure for DexSkin to mitigate these issues and enable policy transfer across hardware instances.

**Calibration procedure.** We develop two calibration protocols. First, DexSkin is enclosed in a 3D-printed airtight chamber with an Ecoflex 00-50 membrane, and internal pressure is ramped from 0–6 psi to impose uniform stress; fitting each taxel's response to an exponential curve provides both a forward mapping ($\Delta C/C_0 \rightarrow$ pressure) and an inverse mapping to align new data with legacy outputs. Second, DexSkin is mounted on a motorized stage with a force gauge, recording three loading–unloading cycles per taxel to map outputs to normal force.

**Policy transfer experiment.** We revisit the perturbed pen reorientation task using a policy trained on one sensor pair (*Source*) and transferred to swapped fingers (*Target*), simulating sensor replacement. Results (Table 2) show that DexSkin transfers reasonably well without calibration and improves further with it, while DIGIT policies fail entirely when gel windows are swapped due to appearance changes.

| Sensor configuration | Stage 1 | Stage 2 |
|---|---|---|
| Source sensors | 20/20 | 20/20 |
| Target sensors (no calib.) | 17/20 | 12/20 |
| Target sensors (calib.) | 18/20 | 16/20 |
| Source sensors (DIGIT) | 20/20 | 0/20 |
| Target sensors (DIGIT) | 0/20 | 0/20 |

Table 2: Pen reorientation policy performance when transferred across sensor hardware. We report successes across two stages: (1) successfully reorienting the pen the first time and (2) detecting and fixing human perturbation.

## 4.3 Real-World Online Robot Learning with DexSkin

Beyond imitation learning, tactile sensing can provide reward signals for RL, but many sensors lack durability and clear reward definitions. We evaluate DexSkin on a delicate blueberry grasping task (where a single excessive force causes failure), augmenting a non-tactile imitation policy with DexSkin and training a residual policy via soft actor-critic [37, 38], with rewards penalizing excessive forces, large actions, and failed grasps.

Figure 4 shows that the residual policy refines the base actions to produce gentler grasps. Without tactile input, the base policy crushes berries, while the DexSkin-enhanced policy handles them successfully. This demonstrates that DexSkin can adapt policies trained without tactile data and provide natural reward signals without extra classifiers or calibration.

## 5 Conclusion

We present DexSkin, a soft tactile sensor that conforms to varied geometries for localized, high-coverage sensing. Integrated on a gripper, DexSkin improves imitation learning for contact-rich tasks, enables calibration-based transfer across hardware, and provides reward feedback in real-world reinforcement learning, advancing toward practical tactile sensors with human-skin-like coverage and sensitivity.

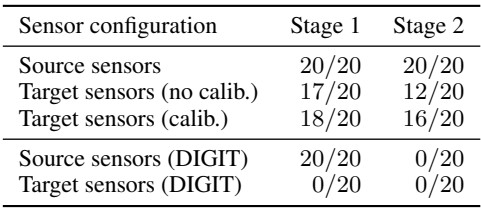

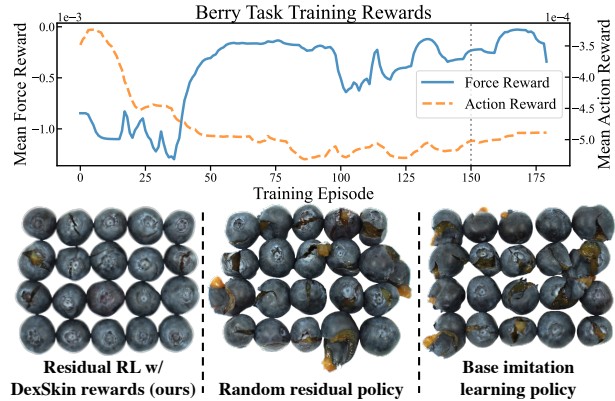

Figure 4: (Top) Residual policy learning curves showing reduced DexSkin-output forces above threshold (*force reward*) and balanced action costs (*action reward*). Training used a faux berry until episode 150, then real berries for 30 episodes. (Bottom) Berries after grasp and transport; the finetuned residual policy causes much less damage.

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
