# OpenReview forum: "DexSkin: High-Coverage Conformable Robotic Skin for Learning Contact-Rich Manipulation"
_robot-learning.org/CoRL/2025/Workshop/Dexterous_Manipulation — CoRL 2025 Workshop Dexterous Manipulation Spotlight_

### Official Review · Reviewer_d265 · 2025-09-07
**Review of DexSkin: High-Coverage Conformable Robotic Skin for Learning Contact-Rich Manipulation**

**Rating:** 7
**Confidence:** 3

**Review:**

### Summary
This paper introduces **DexSkin**, a soft, conformable capacitive electronic skin designed to provide high tactile coverage and sensitive, localized contact sensing for robotic manipulation. DexSkin is demonstrated on parallel-jaw gripper fingers, offering nearly full-surface tactile coverage. The paper evaluates DexSkin in several **contact-rich manipulation tasks**, such as in-hand object reorientation and elastic band wrapping, within a learning-from-demonstration framework. Importantly, the authors show that DexSkin can be **calibrated across sensor instances**, facilitating model transferability, and demonstrate its use in **online reinforcement learning** for real-world manipulation. The results highlight DexSkin’s potential as a practical and scalable tactile sensing solution.

---

### Strengths
- **Clear motivation**: Addresses the long-standing challenge of replicating human-like skin sensing for dexterous robotic manipulation.
- **Technical novelty**: Introduces a high-coverage, soft, and conformable capacitive skin that is adaptable to varied geometries.
- **Comprehensive evaluation**: Includes manipulation benchmarks, calibration experiments, and online RL demonstrations.
- **Practical relevance**: Calibration and transferability make the system suitable for real-world data-driven robotics applications.
- **Strong significance**: Directly advances tactile sensing for dexterous manipulation, an important direction in robotics.

---

### Weaknesses
- **Limited comparison**: The evaluation would be stronger with direct quantitative comparisons against other tactile skins or sensing modalities.
- **Scalability questions**: While demonstrated on parallel-jaw grippers, the challenges of scaling DexSkin to multi-fingered or high-DoF hands are not discussed.
- **Durability and robustness**: Long-term reliability under wear-and-tear or harsh manipulation conditions is not evaluated.
- **Learning results scope**: RL demonstrations are promising but limited in scale; more complex or longer-horizon manipulation tasks could better showcase the benefits.
- **Implementation details**: Some technical specifics (e.g., calibration pipeline, noise handling, sensor response time) are described at a high level and may be difficult to reproduce.

---

### Official Review · Reviewer_gFT2 · 2025-09-10
**Review of the paper: DexSkin: High-Coverage Conformable Robotic Skin for Learning Contact-Rich Manipulation**

**Rating:** 3
**Confidence:** 3

**Review:**

## Overview
This paper presents DexSkin, a conformable capacitive tactile skin that provides localized sensing and adapts to different geometries with human finger-like shape and coverage. Through empirical evaluations, DexSkin is shown to be effective in learning real-world contact-rich manipulation tasks.

## Pros
- Tackling an important problem (manufacturing tactile sensor for robot learning) in the field
- Shows that the tactile data from DexSkin helps the BC policy for learning contact-rich tasks
- High-quality videos effectively demonstrating how the sensor works

## Cons
- This work has been accepted to the main track of CoRL 2025. In accordance with CoRL's established policy discouraging main track papers from simultaneous workshop submissions, this submission may not be appropriate for workshop consideration.

## Recommendation
While this research presents valuable contributions to tactile sensing for robotics, the concurrent acceptance at CoRL 2025's main track creates a conflict with the conference's submission policies. Therefore, I recommend declining this workshop submission despite the technical merit of the work.

---

### Decision · Program_Chairs · 2025-09-18

Accept (Spotlight)